# A Nutritional Investigation of Major Feed Types and Feed Rations Used in Medium-Scale Dairy Production Systems in Sri Lanka

**DOI:** 10.3390/ani12182391

**Published:** 2022-09-13

**Authors:** Sagara N. Kumara, Tim J. Parkinson, Richard A. Laven, Garry C. Waghorn, Anil Pushpakumara, Daniel J. Donaghy

**Affiliations:** 1School of Agriculture and Environment, Massey University, Private Bag 11-222, Palmerston North 4410, New Zealand; 2Department of Farm Animal Production and Health, Faculty of Veterinary Medicine and Animal Science, University of Peradeniya, Peradeniya 20400, Sri Lanka; 3School of Veterinary Science, Massey University, Palmerston North 4410, New Zealand

**Keywords:** cows, forages, energy, protein, tropical dairy farming

## Abstract

**Simple Summary:**

Little information is available regarding the feeding of dairy cattle in Sri Lanka or the impact of feeding on productivity. The aim of this study was to catalogue the availability, quantity, and composition of feeds, to identify the feeding regimens used, and to calculate dietary metabolisable energy (ME) and crude protein (CP) in order to investigate shortfalls in dietary requirements. The ME and CP contents of the abundantly used forages (representing > 50% of the cow diet) Guinea ecotype A and Hybrid Napier CO-3 grasses were generally low (7.5–8.0 MJ/kg DM, 8.0–8.8% DM, respectively), and were lower than that of legumes (i.e., Gliricidia: 10.0 MJ/kg DM, 17.7% DM, respectively). Daily ME intake was consistently 10% lower than the calculated daily energy requirements as a consequence of the low nutritive values of these forages and of farmers’ consistent overestimation of their quality. The CP intake of lactating cows (13.5% DM) was inadequate to meet their requirements (16–18.5% DM), whereas the CP intake of dry cows (11.8% DM) adequately met requirements (11–12% DM). Based on the results of this study, limitations on nutritional requirements adversely affect milk production of dairy cows in Sri Lanka.

**Abstract:**

In this paper, the nutritional quality, digestibility, and chemical composition of major feed types as well as the use of those feeds in rations by medium-scale dairy farmers in the Kurunegala district of Sri Lanka were studied. Nine dairy farms were visited fortnightly over a five-month period to identify the feeds that were commonly used. All farms operated under a stall-feeding system in which a manually mixed ration (MMR) was fed 2–3 times daily. Four forages were identified: Guinea grass ecotype A (*Panicum maximum*), called Guinea grass; Hybrid Napier CO-3 (*Pennisetum purpureum* × *Pennisetum americanum*), called CO-3 grass; Gliricidia (*Gliricidia sepium*); and maize stover (*Zea mays* L.), along with three other supplementary feeds (maize silage, barley distillers’ by-products, and commercially formulated cattle feed). These feeds were subjected to proximate analysis and in vitro digestibility analysis. The metabolisable energy (ME) of the forages ranged from 7.5–10.0 MJ/kg dry matter (DM), with the ME of Guinea grass and CO-3 grass (7.5 and 8.0 MJ/kg DM, respectively) being lower than that of Gliricidia (10.0 MJ/kg DM). The neutral detergent fibre (NDF) concentration of both Guinea grass and CO-3 grass (both 72% DM) was much higher than that of Gliricidia (47% DM). Crude protein (CP) was higher in Gliricidia (17.5% DM) than in either Guinea grass or CO-3 grass (8.0 and 8.8% DM, respectively). The ME of the supplementary feeds varied between 11.0 and 12.8 MJ/kg DM, while CP varied between 15.0 and 24.0% DM. The daily ME intake of cows was consistently 10% lower than their calculated daily energy requirement; for dry cows, the mean intake was 90 MJ/cow/day supplied vs. 101 MJ required, while for cows in early lactation the mean intake was 126 MJ/cow/day supplied vs. 140 MJ required. The average CP intake of lactating cows (13.5% DM) was inadequate (requirements: 16 to 17.5% DM), while the average CP intake of dry cows (11.8% DM) was satisfactory (requirements: 11 to 12% DM). The current study shows that the majority of the feed types used in these medium-scale dairy farms provide insufficient ME or CP to meet the nutritional requirements of either lactating or dry cows irrespective of the quantity of feed provided.

## 1. Introduction

Rapidly increasing population and burgeoning per capita demand for animal protein in Sri Lanka has resulted in escalating demand for livestock products, especially dairy products [1]. Most dairy farmers (70%) in Sri Lanka run small-scale dairy operations (1–10 cows) [2,3], although medium-scale dairying (11–100 cows) [4] is becoming more common. Medium-scale dairy operations are generally conducted under a more intensive feeding system (i.e., higher reliance on concentrate feed and use of cultivated forage crops) than small-scale farms. Nonetheless, although farmers may grow forages themselves, the majority depend on forages harvested from roadsides, paddy fields, and crop lands [5].

In terms of forage quality, energy density is the most crucial factor for milk production, although protein, vitamins, and minerals are important as well [6]. The supply of a balanced diet has an impact on the productivity, welfare, and environmental sustainability of dairy farming [7,8,9]. The quality of harvested forages compared to other available feed resources (animal and/or plant by-products, formulated concentrates, silage, etc.) varies considerably, as all are subjected to different climatic and management practices [10]. For example, the quality of harvested forages is generally low, characterised by high neutral detergent fibre (NDF; >60% on a dry matter (DM) basis), low digestibility (<50%), low metabolisable energy (ME; 6–9 MJ/kg DM), and low concentrations of soluble sugars and starches (<100 g/kg). Management practices such as regular addition of fertiliser, watering, and appropriate harvest intervals, all of which affect forage quality [11,12], are not widely known or implemented in medium-scale dairy systems in Sri Lanka. As forages are the main component of the diet, forage quality has a significant impact upon the ability of the diet to meet the nutrient requirements of cows at different stages of the lactation cycle.

Kurunegala district is the biggest district in the intermediate zone and the main milk-producing region of Sri Lanka [3]. Feeding dairy cows with off-farm fodder grasses is popular among its dairy farmers; however, priority is given to the quantity that is fed, with little attention paid to the quality of the feed [13]. Only a small range of tropical forages are commonly used as dairy feeds. These include Guinea grass ecotype A (*Panicum maximum*, hereafter referred to as ‘Guinea grass’), Hybrid Napier CO-3 (*Pennisetum purpureum* × *P. americanum*, hereafter referred to as ‘CO-3 grass’), Gliricidia (*Gliricidia sepium*) and maize stover (*Zea mays*.; i.e., the residual plant mass after harvesting of the cobs/grain). Guinea grass is a fast growing, leafy, and quite hardy perennial grass that is suitable for a range of climates [14]. It has two types, a tall/medium tussock (>1.5 m at flowering, 1.5–3.5 m tall) and a short tussock (<1.5 m at flowering, 0.5–1.5 m tall) [15]. Guinea grass ecotype A is the shorter type and the most common forage source for farmers in Sri Lanka, as it can be easily harvested along roadsides and railway lines and from natural grasslands and/or scrubland at low and middle elevations [5]. CO-3 grass has become the second most widely distributed forage type, as its overall yield, crude protein (CP), and ME levels are regarded as being higher than many other tropical forage types [5]. A number of development projects (e.g., the Livestock Breeding Project under the Ministry of Agriculture and Livestock) have provided support and training to small holder farmers in the management of CO-3 grass to mitigate the risk of feed shortages during the dry season or when outsourced feed is unavailable. Gliricidia is a leguminous tree that is a useful forage for a dairy production system and can be used as an alternative protein source to replace more costly concentrate feeds [16].

The nutritive value of feeds can be measured in different ways, and validation of values is crucial in situations where there are only limited feed libraries available for a given forage type. In vivo digestibility analysis together with proximate analysis have been carried out in the past to determine the chemical composition of feed types; however, these techniques have been partially replaced with in vitro digestibility studies, which represent a cheap, rapid, and cost-effective laboratory method [17]. In most developed countries, Near-Infrared Spectroscopy (NIRS) is commonly used for feed analysis, as it is more accurate and cost effective than most other methods [18]. However, this method is not commonly available in developing countries, and indeed even proximate analysis and other ‘wet’ chemical methods are often not readily available. Consequently, it has been exceedingly difficult to develop feed bank data for the tropical forages that are commonly used in Sri Lanka.

There is currently limited information about the use or nutrient composition of forages, supplementary feeds, and concentrate types in medium-scale dairy farming in Sri Lanka. Therefore, the main objective of the current study was to identify the forages and supplementary feeds that are most commonly available in medium-scale dairying and to investigate the ME and CP content of those feeds. The second objective of the study was to investigate the total energy and protein supplied through manually mixed rations (MMR), i.e., diets manually prepared by mixing chopped forages and supplements, fed to dairy cows, and to determine whether these meet calculated requirements for dairy cows according to their lactation stage.

## 2. Materials and Methods

### 2.1. Animals and Farms

The study was conducted in the Kurunegala district (7.48° N, 80.36° E), Sri Lanka. The common feature of the rainfall pattern in the Kurunegala district is bimodal, with a long rainy season from March to July, a short rainy season from September to November, and a continuous dry season from December to February [19]. During the year the temperature typically varies from 21 °C to 34 °C and the relative humidity varies from 72% to 83% [19]. Nine medium-scale (11–100 dairy cows) dairy farms were selected on the basis of management type (stall-feeding), feeding practice (MMR), housing system (loose barn), breed(s) (Jersey, Jersey × Holstein-Friesian, and Jersey × Sahiwal), and farming experience (>2 years). The average body weight was 416 ± 8 kg (mean ± standard error of mean (SEM)). Cows were fed 2–3 times daily. A total of 398 cows were enrolled into the study from these farms. Farms were visited every two weeks from May to September 2018.

Each of the farms had a year-round calving pattern [20], and at each visit the cows were classified into dry (up to 14 days before calving), fresh (calving to 30 days in milk), early lactation (31–100 days in milk), mid-lactation (101–200 days in milk), and late lactation (201–300 days in milk) in order to estimate their intake of dry matter (DMI), CP and ME.

Each farmer was asked to provide a list of forages and supplements that they regularly used. From these lists, four forages (Guinea grass, CO-3 grass, Gliricidia, and maize stover) along with three other feeds (maize silage, barley distillers’ byproducts, and commercially formulated cattle feed) were identified for further examination. 

### 2.2. Forages and Supplements

Triplicate samples of the aforementioned forages and supplements were collected at two-month intervals from each of the nine farms. At each sampling event, a representative sample of each forage type was collected from a harvested bulk supply. These were then sub-sampled to obtain a smaller representative portion for chemical composition analysis. Samples of supplementary feeds were randomly collected for chemical composition analysis from bulk stores.

Representative samples of forages were placed on a flat surface for measurements of leaf length, leaf count (live and dead), and stem diameter at the base. Wet weights of the forages were then determined. Forages were chopped into 2–4 cm long pieces and oven dried at 60 °C for 2–3 days until they reached a constant weight. Dried samples were milled into 1 mm particles using a Thomas Hammer Mill [21] at the Veterinary Research Institute (VRI), Sri Lanka. Ground samples were then sent to the Alltech laboratory, Bangalore, India and VRI, Sri Lanka for in vitro digestibility (IVD) assays and CP estimation, respectively.

### 2.3. Feed Nutritive Characteristics

The DM percentage of all feed types was calculated as the difference between the wet (prior to chopping) and dry (after oven drying) weights. Because feed ingredients and dietary proportions in MMR differed from farm to farm the amount of fresh matter of each feed type fed to cows was recorded. These data were used to calculate the total DM offered at the different stages of the lactation cycle and then to estimate the total DMI of each cow. 

Total nitrogen (N) was determined by the Kjeldahl method (AOAC 984.13) [22,23], from which CP was calculated as N × 6.25. The results were recorded as % CP on a DM basis. Ash and ether extract (EE) were determined by the AOAC 938.08 and AOAC 945.16 methods, respectively. The NDF and acid detergent fibre (ADF) were measured by the filter bag technique as described by Tilley and Terry [24].

### 2.4. Feed Digestibility

#### 2.4.1. In Vitro True Dry Matter Digestibility

The true DM digestibility (TDMD) was measured using the Daisy^II^ Incubator in vitro technology [25]. In vitro DM digestibility was determined for all seven feed types in triplicate in an artificial rumen (DaisyII incubator; Ankom Technology^®^, Macedon, NY, USA) following the approach of Tilley and Terry [24] as modified by Goering and Van Soest [26]. The artificial rumen consisted of a thermostatic chamber (maintained at 39 °C) with four rotating jars.

#### 2.4.2. In Vitro Neutral Detergent Fibre Digestibility

The NDF digestibility (NDFD) was estimated using the following formula:$$\%\;\mathrm{NDFD}\;=\frac{100-\left({\mathrm{aNDF}}_{\mathrm{feed}}-{\;\mathrm{aNDF}}_{\mathrm{res}}\;\right)\times 100}{{\mathrm{aNDF}}_{\mathrm{feed}}\;}$$
where aNDF_feed_ = amount (g) of NDF incubated and aNDF_res_ = amount (g) of NDF measured on the residue of fermentation

#### 2.4.3. Apparent Dry Matter Digestibility 

The apparent DM digestibility (ADMD), commonly known as DM digestibility (DMD), was calculated as TDMD minus microbial biomass. The microbial biomass (% DM) for dairy cows under a total mixed ration management system was estimated using ADMD and TDMD data published by Sherasia et al. [27]; thus, the optimum inclusion level of microbial biomass was detected as 5% DM for better digestibility. Therefore, ADMD was estimated using the following formula:ADMD = TDMD − 5 

The ME (MJ/kg DM) of forages and supplements was derived using the calculated ADMD data and estimated EE values. The following formulas were used to calculate the ME [28]:Forages: ME = 0.172 DMD − 1.71
Supplement: ME = 0.134 DMD + 0.235 EE + 1.23 

### 2.5. Estimation of DM, ME, CP Intakes of Dry and Lactating Cows

The mean fresh matter intake (per cow per day) was measured to calculate the mean cow DMI. The DMI of each group of cows (dry and lactating) were separately calculated and averaged to provide the group individual DMI. The mean cow ME and CP intakes were calculated as outlined below:ME intake_(per cow)_ = P_n=1_ × DMI × ME_n=1 +_ … _+_ P_n_ × DMI × ME_n_

CP intake_(per cow)_ = P_n=1_ × DMI × CP_n=1 +_ … _+_ P_n_ × DMI × CP_n_

where, P = Ingredient proportion in the MMR, n = ingredient, DMI = Dry matter intake, ME = Metabolisable energy concentration of the ingredient, CP = Crude protein concentration of the ingredient.

### 2.6. Calculation of Nutritional Requirements of Dry and Lactating Cows

The nutritional guidelines published by NRC [29] and Moran [30] were used to calculate ME, CP, and DMI requirements of dairy cows at different stages of their lactation. The energy partition for maintenance, milk production, pregnancy, body condition, activity, and climatic stress were calculated to assess the total ME requirements. Dairy cows in the present study were managed in free stalls with zero grazing; therefore, energy partitioning for grazing and walking were not considered for the calculation of total energy requirements.

The average body weight (kg), milk fat (%), protein (%), and lactose (%) used for energy calculations were 450, 4.6, 3.6, and 4.85, respectively. The average milk production at fresh, early, mid-, and late lactation were 15, 17, 12, and 10 L/cow/day, respectively. 

### 2.7. Statistical Analyses

The mean values and standard deviation (SD) for proximate chemical components, predicted chemical components, predicted ME, and the measures of digestibility were calculated using a Microsoft Excel 2016 (Microsoft Corp., Redmond, WA, USA) spreadsheet.

## 3. Results

### 3.1. Forage Characteristics

The descriptive parameters of harvested forages, including length, leaf count (live and dead), and stem diameter, are shown in Table 1. The results indicate that CO-3 grass has the longest stems (up to 220 cm), while maize forage has the shortest stems (70–75 cm). The average cutting interval was only reported for CO-3 grass and maize forages. The number of dead leaves recorded for maize stover was high (>40%), as dairy farmers were presented with maize forage from which the cobs had been harvested. Dead material was recorded as between 10–20% for CO-3 and Guinea grasses. The stem diameter averaged 18.3 ± 2.1, 15.8 ± 1.3, and 7.3 ± 0.6 mm in maize stover, CO-3 grass, and Guinea grass, respectively. The average length of CO-3 grass and maize stover increased with cutting interval, and stem diameters increased with average length (Figure 1 and Figure 2).

The chemical composition and in vitro digestibility parameters of all forage species are presented in Table 2 and Table 3, respectively. Gliricidia had more CP (17.67% DM) and EE (3.60% DM) than other grass species. The CP content of fodder grasses was very low (8–8.8% DM). The fodder grasses were all higher in NDF than Gliricidia (>64% DM vs. 47% DM, respectively). Additionally, the NDFD of fodder grasses was higher (>37% NDF) than Gliricidia (25.7% NDF). The TDMD values of all forages were similar and of average to good quality at around 60%. With respect to the ME of forages, Guinea grass had the lowest value, at 7.5 MJ/kg DM, and Gliricidia had the highest value at 10.0 MJ/kg DM. Among the feed sources examined other than the forages, formulated cattle feed had the highest ME level (12.8 MJ/kg DM). The ME of maize silage and barley distillers’ by-products were both ~11 MJ/kg. 

### 3.2. Feed Supply and Requirements

The diets that were fed to lactating and dry cows on each of the nine farms are presented in Table 4. On all farms, lactating cows were additionally provided with a calcium mineral mixture (60–90 g/cow/day). Guinea and CO-3 grasses were the main fodder species used in all diet formulations, and the proportions fed of these forages ranged from 22–48% and 12–60%, respectively. Cows were fed with freshly harvested and/or ensiled mature maize stover. As leguminous trees such as Gliricidia are not abundant in the Kurunegala district, Gliricidia was included in the diet of dairy cows to a maximum of 23% at all lactation stages. Barley distillers’ by-products and formulated cattle feed were used to balance the ME and CP levels of the final diet. The majority of the farmers used a high amount of supplements, in ratios of around 40% compared with 60% forages.

The calculated nutritional requirements/recommendations for dairy cows based on Moran [30] and NRC [29] are presented in Table 5. Estimated DM, CP, and ME intakes are then compared to those recommendations; see Table 6. Based on NRC requirements [29], cows in all stages of lactation experienced a shortage of CP while dry cows received adequate CP. Dry cows and lactating cows at all stages experienced low ME intakes according to calculations based on Moran [30]; however, fresh cows and cows in late lactation had an excess intake of ME according to NRC [29]. The DMI of cows in the dry and lactating stages were lower than recommended by Moran [30]. However, based on NRC [29], the DMI of transition and late lactation cows exceeded recommendations, while the DMI of early and mid-lactation cows was lower than recommended. 

## 4. Discussion

The present study represents the first time that the feeding practices for dairy cows on medium-scale dairy farms in Sri Lanka have been examined with respect to the nutritive values of the available forage and non-forage feeds and the ability of those diets to meet the nutritional demands of late dry and early to late lactation dairy cows. 

Forages are either acquired on an opportunistic basis (e.g., cut and carted from verges; collected from post-harvest maize stems) or are intentionally grown as feeds. Farmers have more control over the feed characteristics of the forages that they grow themselves through management of the cutting cycle, although the ME and CP of forages is generally low and the NDF is high. Significant contributions to energy come from supplementary feeds (e.g., brewers’ grains, maize silage) and proprietary/formulated cattle compound feeds. However, even with these supplementary feeds, we found that the diet was unable to meet the full dietary needs of most cows at most stages of lactation. Hence, perhaps unsurprisingly, the majority of milking cows are primiparous, with multiparous cows being relatively rare. 

### 4.1. Cow Diet Composition, Shortage/Excess of Nutrients

The nutritional requirements for dairy cows in the current study were calculated based on both NRC [29] and Moran [30]. The preferred guideline used to interpret the shortage and/or excess of ME, CP, and DM of given feed rations was that of Moran [30], as it was specifically developed to assess tropical dairy production systems [31,32]. The required energy values for maintenance and pregnancy based on Moran [30] were higher than those recommended by NRC [29] (Table 7), and the calculated total energy requirements for all stages of the lactation were therefore higher than supplied. 

The CP concentration of all forages in the current study was higher than the critical CP intake (8% DM for rumen function) required for unrestricted feed intake [33]. Nonetheless, based on the CP requirements from both Moran [30] and NRC [29], dairy cows in the fresh, early, and mid-lactation stages had CP concentrations (<14% DM) that were lower than recommended (14–18% DM). Low CP is characteristically associated with a negative effect on milk protein production [34], though not milk volume [35]. Broderick [36] reported that by increasing the CP in the diet from 15.1% to 18.4%, milk protein and milk fat yields increased by 3% and 4%, respectively. Sinclair et al. [37] noted that low CP levels had no significant effect on milk production or on animal health or fertility, although their definition of ‘low CP’ (14–15% DM) was higher than the CP values seen in the forages in the current study. There may be a self-perpetuating element in low dietary CP, as dietary CP can affect DMI. Broderick [36] found that increasing dietary CP was associated with an increase in DMI, whilst Kalscheur et al. [35] reported that compared to low CP diets (13%), high CP diets (23%) were associated with an increase in both DMI and body weight. As DMI values in the present study were lower than recommended, it is possible that the cows on the study farms were adversely affected by the low CP diets provided.

Based on the DMI and ME requirements from Moran [30], all cows in the current study, both dry and lactating, experienced deficits of DMI and ME; therefore, there is clear evidence that cows were at a mild to moderate negative energy balance (NEB) during the transition period and that the energy imbalance remained evident until the end of lactation. Inadequate DMI during the transition and early postpartum increases BCS loss and BW loss and reduces both milk production and fertility [38,39].

### 4.2. Forage and Supplement Analyses

The DMD and ME values as well as the CP for the forages in the present study depended upon the proportion of leaf and the maturity of the forage at harvest. The number of leaves per plant is a useful parameter for calculating the growth, DM yield, and nutritive values of fodder species [40]. On the other hand, the number of leaves, leaf length, and plant height of fodder grasses increase with maturity or stage of harvesting [41], and the proportion of green:dead leaf is thus as important as the total leaf mass in determining the nutritive value of the forage. In the present study, CO-3 grass and post-harvest maize stems contained more leaves than Guinea grass, while the number of dead leaves increased with longer cutting intervals for green forages; the lowest green:dead leaf ratio was in maize stover. Waghorn & Clark [42] stated that forage maturation increases fibre content and reduces CP and carbohydrate contents. This reduction in forage quality with maturation results in a decline in the forage digestion rate, CP intake, and DMI of cows [6]. 

In the current study, the stem diameter of harvested CO-3 grass and maize stover increased with the stem length (Figure 2), and for both grasses the stem length increased with the cutting interval (Figure 1). These results are consistent with the findings of Wangchuk et al. [43], who reported that the basal circumference of Napier grass is positively correlated with its height and height is similarly positively correlated with cutting interval, with an average height of 151, 218, and 256 cm being seen at 40, 60, and 80 day cutting intervals, respectively. Height is a critical factor in quality; Bernard et al. [44] reported that the height to which maize is allowed to grow significantly influences the nutrient density, nutrient digestibility, and DMI of dairy cows. In addition, letting plants grow has long term consequences for future quality. Orodho [45] reported that cutting CO-3 grass to a low level (i.e., leaving a 10–15 cm stump) positively influences subsequent yield quality and plant regrowth. In other words, while letting forage grasses grow tall and cutting them low may maximise the mass collected at that harvesting, it impairs both the nutritive value of the harvested material and the subsequent regrowth of the crop [46]. The practical implications of the results from the current study, backed up by the historical literature, are that Sri Lankan dairy farmers should select forages containing a high proportion of live leaves and a minimum of stem.

The total NDF concentration and NDFD of the forages are major factors in determining forage quality [47]. The high NDF concentration (>70% DM) of the Guinea and CO-3 grasses in the current study resulted in low DMI during the dry and mid-lactating periods. The effects of advancing maturity upon forage digestibility and concurrent adverse effects upon DMI have been described by Ball et al. [6], by Waghorn and Clark [42], and by Allen [48], showing that increased NDF concentration in the final diet significantly reduces DMI and milk production. Whether this decrease in DMI is solely due to the concentration of NDF is perhaps debatable, as Jung & Allen [49] concluded that rumen fill, and hence DMI, is affected by factors such as NDFD, particle size, and the chemical composition of the feed, while Oba and Allen [47] found that the DMI and milk production of dairy cows decrease with low NDFD of forages. In the current study, the NDFD was low at <45% in major forages (Gliricidia: 25.7%, Guinea grass: 37.7%, CO-3 grass: 43.3%), which likely contributed to the reduced DMI. 

Due to the lack of land availability, growing of maize was considered to not be economically feasible for dairy farmers. In contrast, buying the mature forage remaining after sweet corn harvest was feasible, as it is abundantly available, and therefore cheap, during the sweet corn harvesting season. However, the nutritive value of maize declines markedly from the wet to the post-harvest dry stages [50] in terms of higher NDF and ADF, and thus the practice of using post-harvest maize forages undoubtedly affects the quality of the cows’ diet. As an aside, there has been debate about the NDFD of the legume Gliricidia. Hoffman et al. [51] reported that legumes generally have low NDF concentrations and low NDFD compared to grasses, whereas Aregheore et al. [52] reported a far higher NDFD (43.9%) for Gliricidia. It is possible that the lower NDFD value obtained in the current study was due to the use of more mature Gliricidia leaves and stems, as digestibility decreases with maturity of forages [51], although no data were collected in the present study regarding the maturity of Gliricidia.

The CP concentration of Guinea grass (8.0 ± 0.3% DM) and maize stover (8.7 ± 0.8% DM) were similar to values that have been previously reported [5,41,53], although the CP concentration of CO-3 grass (8.8 ± 0.7% DM) was lower than the values reported by Weerasinghe [5] and Pavithra et al. [53], which might be due to either location-specific variations and/or differences in the maturity of the forage. The CP concentration of Gliricidia was twice that of fodder grasses, which was expected as it is a legume. The CP concentrations of formulated cattle feed and barley distillers’ byproducts were double and triple that of maize silage, respectively; however, low inclusion (<15% DM) of these feeds in the overall ration meant that their net contribution to total dietary CP was relatively limited. 

The fat concentration (EE) of all forages along with silage and supplementary feeds ranged between 1.9 and 7.2% DM. The fat content of fodder grasses in the current study ranged from 1.9 to 3.6% DM which is comparable to the values recorded by Warly et al. [54] for tropical grasses (2.7–3.9% DM). The fat concentration of maize stover was the lowest (1.9% DM) because mature maize stems contain very low amounts of fatty acid [55].

### 4.3. In Vitro Digestibility and ME Content of Forages and Supplementary Feeds

The DMD for forages ranged from 51–65% DM, which is similar to the range reported by Warly et al. (2004) (49.9–62.2). Within the fodder species, Gliricidia had the highest TDMD, ADMD, and ME values, which were consistent with published values [14], As expected, the highest TDMD and ADMD overall were reported for the formulated cattle feed.

With respect to the ME values of forages, both Guinea and CO-3 grasses contributed the least ME to the diet across all stages of lactation. The use of a high proportion of both these forage types (15–45% of Guinea grass and 30–60% of CO-3 grass) in the diet of cows in the current study resulted in an energy shortage during the dry period and mid-lactation. Farmers usually feed the highest level of supplements during early lactation due to a high milk response, then less later in lactation and during the dry period. However, poor nutrition and/or a lack of supplementation during early to mid-lactation may result in low body condition gain, which has a negative effect on milk production. Furthermore, the level of energy intake during the period immediately pre-calving can have a significant influence on post-calving metabolic diseases, milk production [56,57], and fertility [58].

## 5. Conclusions

When the ME, CP, and DMI requirements for different stages of lactation were calculated, dry cows and mid-lactating cows were shown to have a shortage of ME and DMI, and all lactating cows had a shortage of CP. Overall, the quality of Guinea and CO-3 grasses was low compared to the leguminous tree (Gliricidia) and maize stover. Ensiling maize resulted in ME values of 11 MJ/kg DM, with an NDF value of 52% DM and NDFD value of 50.7% NDF, which are all adequate to maintain milk production. The DMD of forages was higher than the minimum required to be classified as ‘good quality’ forage; however, the NDFD of Gliricidia was below the required minimum. Alternative feeds, such as barley distillers’ byproducts and formulated cattle feeds, are higher quality supplements that provide a comparatively higher amount of ME and CP for the final diet, although are more costly than tropical forages.

Overall, it is therefore clear that the forage component of the diets of the cows in this study was the key limitation on productivity. Improvements to the time of harvesting grasses would clearly improve ME as well as digestibility and CP content, and because of the limiting effect of these upon DMI, an overall increase in DMI (and hence of the key energy and protein indicators of the diet) would probably eventuate. While the limitations imposed by poor quality forages can be ameliorated to an extent by higher quality supplementary feeds, it is clear that the key to improving the productivity of Sri Lankan dairy cows lies primarily in management of the quality and quantity of forages. 

## Figures and Tables

**Figure 1 animals-12-02391-f001:**
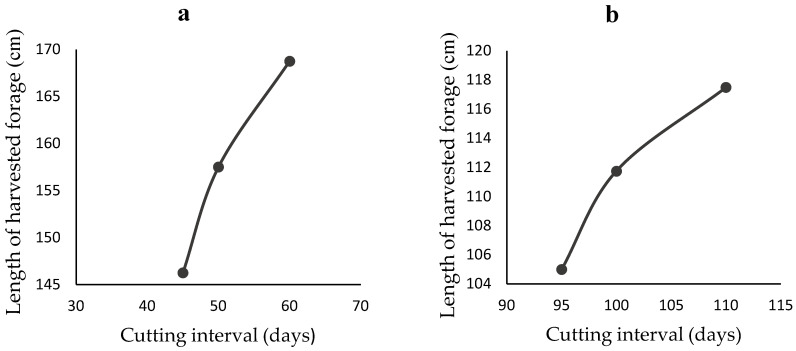
Change in average length of harvested forages over the cutting intervals: (**a**) CO-3 grass (n = 3) and (**b**) Maize (n = 3).

**Figure 2 animals-12-02391-f002:**
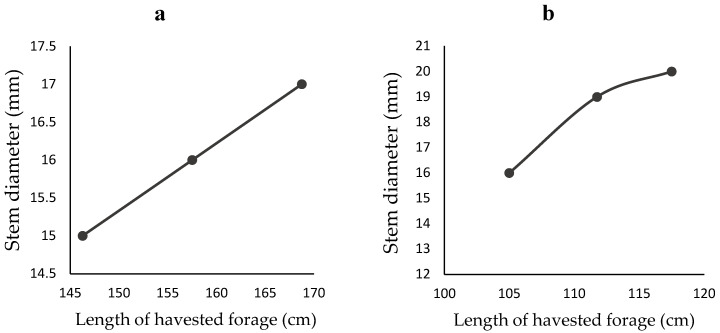
Average change in stem diameter over length of harvested forages: (**a**) CO-3 grass (n = 3) and (**b**) Maize (n = 3).

**Table 1 animals-12-02391-t001:** Harvested length (mean ± standard deviation (SD), range), leaf count (green, dead), stem diameter (mean ± SD, range), and presence of inflorescence of forages used in the selected nine dairy farms in the Kurunegala district, Sri Lanka.

Form	Cutting Interval (Days)	Length of Harvested Forage (cm)		Leaf Count (n)		Stem Diameter (mm)		Presence of Inflorescence
		Mean (SD)	Range	Green (Range)	Dead (Range)	Mean (SD)	Range	
Guinea grass								
Fresh	N/A	116 ± 7.0	75–155	2–5	1–2	7.3 ± 0.6	4–10	No
CO-3 grass								
Fresh	45-60	158 ± 11.3	110–220	9–12	1–3	15.8 ± 1.3	13–20	No
Gliricidia								
Fresh	N/A	N/A	N/A	N/A	N/A	13.3 ± 2.5	10–16	No
Maize stover								
Harvested	95-110	123 ± 3.9	70–165	4–6	6–8	18.3 ± 2.1	14–22	Yes

N/A: not available.

**Table 2 animals-12-02391-t002:** Dry matter (DM; % as fed), ash, crude protein (CP; % DM), ether extract (EE; % DM), neutral detergent fibre (NDF; % DM), and acid detergent fibre (ADF; % DM) composition (mean ± SD) of feed sources available at the medium-scale dairy farms in the Kurunegala district, Sri Lanka.

Feedstuff	DM % as Fed	Ash (% DM)	CP (% DM)	EE (% DM)	NDF (% DM)	ADF (% DM)
**Forages**						
Guinea Grass	23.6 ± 1.0	11.6 ± 1.4	8.0 ± 0.3	2.2 ± 0.4	71.7 ± 4.3	41.6 ± 2.8
CO-3 grass	18.8 ± 0.9	9.1 ± 0.9	8.8 ± 0.7	2.9 ± 0.6	71.6 ± 4.3	38.4 ± 3.7
Gliricidia	26.0 ± 1.2	10.0 ± 1.0	17.7 ± 1.2	3.6 ± 0.6	47.1 ± 4.4	33.7 ± 4.2
Maize stover	27.7 ± 1.4	8.1 ± 0.1	8.7 ± 0.8	1.9 ± 0.1	65.0 ± 4.0	32.8 ±3.3
**Supplementary feeds**						
Maize silage	29.1 ± 1.8	6.9 ± 0.7	7.5 ± 0.9	2.4 ± 0.2	52.0 ± 2.4	26.7 ± 2.0
Barley distillers’ by-products	26.1 ± 2.0	4.0 ± 0.4	24.2 ± 2.4	7.2 ± 0.9	51.6 ± 5.4	20.5 ± 2.1
Formulated cattle feed	89.0 ± 1.8	8.3 ± 1.2	15.7 ± 1.9	6.0 ± 0.4	28.1 ± 1.4	N/A

N/A: not available.

**Table 3 animals-12-02391-t003:** Neutral detergent fibre digestibility (NDFD; % NDF), true dry matter digestibility (TDMD, % dry matter (DM)), apparent dry matter digestibility (ADMD, % DM), and metabolisable energy (ME; MJ/kg DM) of feed sources available at the medium-scale dairy farms in the Kurunegala district, Sri Lanka.

Feedstuff	NDFD (% NDF)	TDMD (% DM)	ADMD (% DM)	ME ^a^ (MJ/kg DM)	ME ^b^ (MJ/kg DM)	ME ^c^ (MJ/kg DM)
**Forages**						
Guinea Grass	37.7 ± 3.1	58.4 ± 3.4	53.4 ± 3.4	7.5 ± 0.6	7.9 ± 0.7	6.8
CO-3 grass	43.3 ± 2.7	61.2 ± 3.8	56.2 ± 3.8	8.0 ± 0.5	8.2 ± 1.1	7.7
Gliricidia	25.7 ± 4.0	73.0 ± 6.6	68 ± 6.6	10.0 ± 1.1	9.3± 1.6	N/A
Maize stover	45.0 ± 3.1	66.5 ± 4.1	61.5 ± 4.1	8.9 ± 0.5	9.6 ± 0.7	N/A
**Supplementary feeds**						
Maize Silage	50.7 ± 1.6	74.0 ± 1.3	69 ± 1.3	11 ± 0.2	10.8	N/A
Barley distillers’ by-products	36.8 ± 3.1	67.5 ± 4.6	62.5 ± 4.6	11.3 ± 0.4	11.3	N/A
Formulated cattle feed	40.2 ± 3.9	81.0 ± 3.3	76 ± 3.3	12.8 ± 0.4	N/A	N/A

Values from in vitro digestibility are expressed as mean ± SD (standard deviation); ^a^ Estimated ME from predicted ADMD and EE values; ^b^ ME values obtained from gas production [14]; ^c^ ME values obtained from the published literature, Sri Lanka [5]; N/A: not available

**Table 4 animals-12-02391-t004:** Components (feedstuff) of the manually mixed rations given to dry and lactating dairy cows at each of the nine dairy farms selected for this study.

Feedstuff	Dairy Farms																	
	1		2		3		4		5		6		7		8		9	
	D	L	D	L	D	L	D	L	D	L	D	L	D	L	D	L	D	L
**Forages**																		
Guinea grass	0.0	0.0	45.5	22.2	47.6	23.3	35.7	23.3	45.5	37.5	32.7	0.0	30.3	23.8	30.3	31.8	0.0	0.0
CO-3 grass	60.6	40.0	30.3	27.8	47.6	46.5	47.6	46.5	30.3	37.5	58.2	40.0	48.5	12.7	48.5	12.7	40.0	40.8
Gliricidia	15.2	13.9	0.0	13.9	0.0	18.6	11.9	23.3	15.2	12.5	0.0	0.0	0.0	0.0	0.0	0.0	0.0	0.0
Maize stover	0.0	10.0	0.0	0.0	0.0	0.0	0.0	0.0	0.0	0.0	0.0	18.0	0.0	12.0	0.0	0.0	40.0	15.1
**Supplementary feeds**																		
Maize silage	0.0	0.0	0.0	0.0	0.0	0.0	0.0	0.0	0.0	0.0	0.0	18.0	0.0	10.8	0.0	14.8	0.0	15.1
Barley distillers’ by-products	15.2	22.2	15.2	22.2	0.0	0.0	0.0	0.0	0.0	0.0	0.0	10.0	9.1	18.1	9.1	18.1	12.0	10.8
Formulated cattle feed	9.1	13.9	9.1	13.9	4.8	11.6	4.8	7.0	9.1	12.5	9.1	14.0	12.1	22.6	12.1	22.6	8.0	18.2

D: dry cows; L: lactating cows.

**Table 5 animals-12-02391-t005:** A comparison of the amount of metabolisable energy (ME; MJ/day), crude protein (CP; % DM), and daily DM intake (DMI; kg DM) of dairy cows in the current study compared to guidelines published by NRC [29] and Moran [30].

Stage	ME (MJ/day)			CP (% DM)			DMI (kg DM)		
	Moran 2005	NRC 2001	Current	Moran 2005	NRC 2001	Current	Moran 2005	NRC 2001	Current
D	101.4	93.7	90.4	10–12	10.8–12.4	11.8	10.1	9.37	9.84
F	126.4	112.1	125.6	16–18	15.9–18.7	13.6	12.6	11.2	12.4
E	139.6	138.1	126.1	16–18	16.1–17.6	13.6	13.9	13.8	12.4
M	149.3	128.6	125.6	14–16	16.1–17.6	13.5	14.9	12.9	12.4
L	137.5	121.7	125.6	12–14	13.5–14.8	13.5	13.8	12.2	12.4

D: Dry cows (−14 days to calving); F: Fresh cows (1–30 days in milk); E: Early lactation (31–100 days in milk); M: Mid-lactation (101–200 days in milk); L: Late lactation (201–300 days in milk).

**Table 6 animals-12-02391-t006:** Calculated deficit and/or excess of metabolisable energy (ME; MJ/day), crude protein (CP; % DM), and daily DM intake (DMI; kg DM) of dairy cows in the current study compared to guidelines published by NRC [29] and Moran [30].

Stage	ME (MJ/day)		CP (% DM)		DMI (kg DM)	
	Moran 2005	NRC 2001	Moran 2005	NRC 2001	Moran 2005	NRC 2001
D	−10.9	−3.24	+0.80	+0.2	−0.3	+0.47
F	−0.84	+13.6	−3.42	−3.72	−0.25	+1.18
E	−13.4	−11.9	−3.4	−3.4	−1.53	−1.38
M	−23.8	−3.03	−1.55	−2.55	−2.49	−0.42
L	−11.9	+3.83	+0.45	−0.55	−1.32	+0.26

D: Dry cows (−14 days to calving); F: Fresh cows (1–30 days in milk); E: Early lactation (31–100 days in milk); M: Mid-lactation (101–120 days in milk).

**Table 7 animals-12-02391-t007:** Metabolisable energy (MJ ME/d), protein (% dry matter (DM)), and DM/d (kg) requirements for dairy cows at the late pregnant, fresh, early lactating, mid-lactating, and late lactating stages.

Activity	Late Pregnant (2 Weeks to Calving)		Fresh (Calving to 30 DIM)		Early Lactating (31–100 DIM)		Mid-Lactating (101–200 DIM)		Late Lactating (201–300 DIM)	
	Moran 2005	NRC 2001	Moran 2005	NRC 2001	Moran 2005	NRC 2001	Moran 2005	NRC 2001	Moran 2005	NRC 2001
**ME requirement**										
Maintenance + Activity (MJ/d)	49	44	49	32.7	49	32.7	49	32.7	49	32.7
Pregnancy (MJ/d)	20	19.1	0	0	0	0	0	0	0	0
Milk production (MJ/d)	0	0	88.5	84.1	100.3	90.9	70.8	73.8	59	66.9
BCS (gain/loss) (MJ/d)	27.5	23.9	14	20.15	14	8.25	22	14	22	14
Climatic stress (MJ/d)	4.9	6.54	4.9	6.54	4.9	6.54	4.9	6.54	4.9	6.54
Total ME requirement (MJ/d)	101.4	93.7	128.4	103.2	140.2	138.4	146.7	127.1	134.9	120.2
**Protein requirement (% DM)**	10–12	10.8–12.4	16–18	15.9–18.7	16–18	16.1–17.6	16–18	16.1–17.6	12–14	13.5–14.8
**Daily DM requirement (kg)**										
If feed; 8 MJ/kg DM)	12.7	11.7	16.1	12.9	17.5	17.3	18.3	15.9	16.9	15.1
If feed; 10 MJ/kg DM)	10.1	9.37	12.8	10.3	14.1	13.8	14.7	12.7	13.5	12.1

DIM: days in milk; MJ/d: megajoule/day; % DM: percentage of dry matter; BCS: body condition score; Formulas, values, and assumptions for Moran’s [30] calculations; metabolisable energy loss for activity: zero for inhouse cows, energy for pregnancy: maximum energy utilization at the late pregnancy, BCS: gain (0.5 kg/day) during late pregnancy, mid- and late lactation, loss (0.5 kg/day) during fresh and early lactation, energy for climatic stress: 10% of the maintenance requirement; Formulas, values, and assumptions for NRC [29] calculations; energy requirement for maintenance: 0.33472 MJ/kg body weight, energy loss for activity: zero for inhouse cows, energy for pregnancy: NRC (2–18), energy for milk production: NRC (2–16), BCS: NRC (2–23, 2–24, 2–25, Table 2, Table 3 and Table 4, 2–5), energy for climatic stress: 20% of maintenance requirement; Milk production: Average values of 15, 17, 12, and 10 L/day for fresh, early, mid-, and late lactation were used to calculate energy requirements for milk production; Average body weight: 450 Kg; average fat content: 4.6%; average protein content: 3.6%.

## Data Availability

Data is contained within the article.

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
