# Peer review of "A Nutritional Investigation of Major Feed Types and Feed Rations Used in Medium-Scale Dairy Production Systems in Sri Lanka"

_animals, 2022, doi:10.3390/ani12182391_

Round 1

Reviewer 1 Report

Ms. Ref. No.:  animals-1801735

Title: “A nutritional investigation of major feed types and feed rations used in medium-scale dairy production systems in Sri Lanka

Animals

General comments

I have had the opportunity to review the manuscript. The manuscript is interesting and is in the topic of the journal, however it needs more details in the methodology some points in the discussions

 Below my considerations: 

Introduction:

The introduction is well structured and described, however I suggest reading and adding the following manuscript that can expand and better describe other species: 

https://doi.org/10.3390/ani9050246

https://doi.org/10.3390/ani10030515

Braghieri, A.; Pacelli, C.; Bragaglio, A.; Sabia, E.; Napolitano, F. The Hidden Costs of Livestock Environmental Sustainability: The Case of Podolian Cattle. In The Sustainability of Agro-Food and Natural Resource Systems in the Mediterranean Basin; Vastola, A., Ed.; Springer: Berlin/Heidelberg, Germany, 2015; pp. 47–56.

Materials and methods: more details need to be added regarding:

- location of the geographical area, e.g. latitude longitude;

- climatic references of the area: temperature humidity rainfall;

- line 128 - 152, appropriate references should be added 

- line 221-224 excel is not a suitable software for statistical analysis, process data with appropriate software e.g. SAS

Reviewer 2 Report

The article provides very relevant information. However, the introduction is too long and some key information is missing about the animals and farms. Most importantly, what is the average weight of the animals under the trial from the different herds used in the study?

Line 13 – 25: Simple summary is presented as the abstract. Edit this part of the article in a non-technical way that could easily be interpreted by an ordinary reader.

Line 25: Change Cows were fed 2 3 times “Cows were fed 2-3 times daily.”

Line 136: 60 90 g/cow/day???? 60-90g/cow/day?

Reviewer 3 Report

Dear authors, I recommend the manuscript to be reviewed in order to improve the English vocabulary. In addition, please consider the suggestions listed below. 

Lines 14-16

Please use the same objectives described on lines 112-118 or as suggested below. 

The aim of this study was to first catalogue the availability, quantity, and composition of feeds. Secondly, identify the feeding regimens used for each, and last, calculate dietary ME and CP to investigate shortfalls in dietary requirements. 

Line 16

ME and CP. Please describe it and then use acronyms.

Line 20-25

Daily ME intake was 10% lower than the calculated daily energy requirements, as a consequence of the low nutritive values of these forages, and the farmers’ consistent overestimation of their quality. The CP intake of lactating cows (13.5% DM) was also inadequate to meet daily requirements (16-18.5% DM), whereas the CP intake of dry cows (11.8% DM) adequately met requirements (11-12% DM). In this study, limitation on nutritional requirements significantly affect milk production of dairy cows in Sri Lanka. 

Line 29

Feeds commonly used. 

Line 124 

Breed(s)

Line 125

Fed 2-3 times daily

Line 127

…Every two weeks from May to September 2018

Line 128

Calving patterns and at each visit, cows were… 

Line 131

…To estimate intakes of dry matter (DMI), CP, and ME. 

Line 136

60-90 g/cow/day

Line 139

… Aforementioned forages and supplements were…

Line 259

NDF content 

Lines 514-524

This paragraph could be used in the discussion. Limit your conclusion to the next paragraph. 

Line 526

Key limitation upon productivity

Line 526

Time of harvesting grasses would.. 

Author Response

Dear reviewer,

Round 2

Reviewer 1 Report

Ok, well done.

Reviewer 3 Report

Dear authors, thank you for accepting the suggestions. Good luck with your projects.